# Experimental Study on Seismic Behavior of PC Walls with Alveolar-Type Horizontal Joint under Pseudo-Static Loading

**DOI:** 10.3390/ma15062301

**Published:** 2022-03-20

**Authors:** Junfeng Cheng, Xiaoyong Luo, Laixiu Cheng, Qian Cheng, Linsong Chen

**Affiliations:** 1School of Civil Engineering, Central South University, Changsha 410075, China; chengjunfeng@csu.edu.cn (J.C.); 204812223@csu.edu.cn (Q.C.); lschen@csu.edu.cn (L.C.); 2Engineering Technology Research Center for Prefabricated Construction Industrialization of Hunan Province, Changsha 410075, China; 3Department of Construction Engineering, Gannan University of Science and Technology, Ganzhou 341000, China; chlaixiu@126.com

**Keywords:** precast concrete (PC) walls, pseudo-static loading tests, axial compression ratio, alveolar-type horizontal joint, grouted sleeve connection

## Abstract

There are many horizontal joints on precast concrete (PC) wall panel structures, which certainly has a significant impact on the seismic behavior of structures. This paper proposes a novel alveolar-type horizontal joint, which has advantages of convenient and rapid assembly. Six precast concrete wall specimens with alveolar-type joints were designed and constructed, and they were weakly connected by spliced rebars anchored into grouted sleeves to meet the requirements of structural performance. The pseudo-static loading tests on these specimens were conducted to investigate the effects of influencing factors, such as the axial compression ratio, the thickness of wall (interface contact area), and the addition of a vertical grouted sleeve connection at the horizontal joint, on the seismic performance of PC walls. Analyses and comparisons were conducted in terms of the cracking propagation pattern, failure modes, force–displacement hysteretic curves, skeleton curves, bearing capacity, ductility factors, and energy dissipation of PC walls. It was concluded that the axial compression ratio and adding grouted sleeve connection had a significant influence on the cracking mode of PC walls, whereas the impact of the wall thickness was slight. The shear capacity and energy dissipation capacity of specimen dramatically enhanced by increasing the axial compression ratio or adding grouted sleeve connection. The PC wall exhibits good ductility after adding the vertical grouted sleeve connection at a horizontal joint. However, the ductility factor increases firstly and then decreases in the enhancement of the axial compression ratio. The reduction in wall thickness has remarkable impacts on the shear strength and energy dissipation capacity of specimens, but the influences on ductility were not significant. The prediction method for calculating the shear capacity of PC walls with alveolar-type horizontal joints was proposed based on the experimental data, and these calculated results are in good agreement with the experimental results.

## 1. Introduction

Precast concrete (PC) technologies have made substantial advances in industrial production, facility construction and environmental protection among others, and it has been widely developed and applied all over the world such as Europe, Japan and China, etc. [1,2,3,4]. Some scholars have carried out numerous works on the connection properties of vertical or horizontal joints in PC structures [5,6,7,8], and these connection behaviors are regarded as one of crucial factors affecting the seismic performance and integrity of structures [9]. Precast concrete wall panel structural system is a kind of prefabricated concrete structural system with a higher assembly rate, and it has been largely under development and application in recent decades. Mochizuki [10] and Bhatt [11] discussed the seismic behaviors of PC shear walls with vertical or horizontal joints, and found that the different tectonic forms of horizontal and vertical joints would have significant impacts on the seismic performances, in which the horizontal joint directly affected the ultimate bearing capacity. The steel grouted sleeve connector is one of the typical grouted connectors widely performed in practical engineering, and there are numerous investigations on precast concrete structures connected by the grouted sleeve connectors [12,13,14], including testing investigations [15,16] and theoretical studies [17,18], as well as finite element analysis [16,19]. The reliability and consistency of connection properties on the grouted sleeve connector to transfer longitudinal reinforced bars’ stress were demonstrated at horizontal joint [20,21], which was safe and dependable. Meanwhile, these influence factors, such as the diameter of steel bars, the lapped length of rebars, and the strength and hoop size of reinforced bars, were considered to analyze the connection performances of connector [22,23]. It was concluded that the seismic performances of the precast concrete (PC) wall connected by double-row grouted sleeves were equivalent to that of the cast-in-place wall, such as the bearing capacity, ductility, and energy dissipation capacity, etc. [24]. In addition, different vertical connections for PC walls were also investigated, included loop connection, spiral-confined lap connection [25,26]. and unbonded post-tensioned connection [27]. The primary objective of these PC wall panel structural systems up to now is to realize the goal of “equivalent to cast-in-place” by the reliable connections [25,28,29], based on the American Concrete Institute (ACI) 318 code [30]. However, for the existing PC structure, wet operation on site was still inevitable during the process of wall panels assembly. Moreover, a large amount of longitudinal reinforcement needs to be connected by a connector, and higher precision demands in the assembly process of PC members may cause drawbacks, such as high cost, low installation efficiency, and so on in situ [24,31]. The aforementioned problems may be determined as restricts for the long-term development of PC structures [32].

In recent years, with the advancement and development of technology, many scholars pay more attention to these problems. As mentioned previously, the reinforcement connection can be classified as three types of connection: full connection (double-row), partial connection (single row or staggered layout), and no connection (seat slurry), based on the connection type between the upper and lower layers. For this purpose, some novel connection joint methods were adopted by [24,25,26,27,31]. Liao et al. [24] proposed a new type of monolithic assembled concrete shear wall structure with non-connected vertical distributed reinforcement, based on the load capacity of equal strength with the cast-in-place structure, and found that the proposed shear walls exhibit a similar bearing capacity, ductility, and energy dissipation to the cast-in-place shear wall. Meanwhile, PC walls, in the form of large-diameter and large-spacing rebars or uniformly reinforced rebars, are reported [14,31], and the technologies can ensure easy and economical installation, and thus enhance the assembly speed and effective construction. Furthermore, many researchers also proposed the novel idea of fully precast concrete wall panel structural systems to realize the industrial manufacture and rapid construction for PC structures [33,34,35,36,37,38,39,40,41]. These structural systems included monolithic precast concrete round-hole wall panel structures [33,34,35], precast concrete (PC) frame-shear wall structures [36], and assembled large-plate structures [37,38,39]. It was suggested that the longitudinal bars of the side column should be connected by steel grouted sleeve connectors, and the results illustrated that these PC wall panel structural systems exhibit good seismic performances. Yu and Guo [40,41] proposed the novel low-rise precast concrete wall panel structural system with bolted connections, which has many advantages due to its faster assembly, convenient and reliable connection properties, etc.

To sum up, although there are numerous investigations on the connection properties of vertical joints [12,13,14,15,16,17,18,19] and seismic performances [24,25,26,27,28,29] for precast concrete wall panel structural systems [8,37,38,39], they mainly focus on the high-rise building residential structure. However, there are few reports on precast concrete wall panel structures connected by the steel grouted sleeve connector and applied to low-rise or multi-story residential buildings [42,43]. It is worth mentioning that above 60% of cities and towns regions in China are located at low-intensity fortification zones (6 degree of fortification) or seismic zones below 7 degrees [32]. Besides, the existing assembled construction method and design standards of high-rise buildings directly applied to the low-rise or multi-story PC buildings are obviously wasteful and uneconomical [41]. Meanwhile, the multi-story building has lower requirements on the connection properties of joints; thus, it was suitable to employ simplified joints to form the strong structure and weak connection characteristics, and which meet the seismic performance of a monolithic assembled wall panel structural system, based on the performance-based design of structures [44].

As a result, this paper proposes a novel horizontal joint which is high at both ends and low at the middle, and which has advantages of convenient and rapid assembly, and the section along the horizontal joint presents alveolar-type features, as shown in Figure 1. It not only provides stability by relying on the self-weight of the wall panel, but also helps to prevent water entering from the external wall to the internal wall. The joint made of cement mortar has high strength and flexible properties [45]. Meanwhile, the bottom and top surface of the PC wall were roughened during the process of manufacturing concrete members to enhance the friction or roughness at the interface [43,44,46]. The horizontal joint was weakly connected by several single-row steel grouted sleeve connectors to realize the indirect connection between the vertical walls, which meets the application and rapid construction requirements for the low-rise or multi-story precast concrete wall panel buildings. Furthermore, the influencing factors and seismic behaviors of monolithic precast concrete wall panel structures with steel grouted sleeve connections need to be further discussed [2,45]. Consequently, a total of six precast concrete wall specimens with alveolar-type horizontal joints were designed and constructed, and meanwhile the pseudo-static loading tests on these specimens were conducted to investigate the impacts of crucial factors on the seismic performance of specimens. These factors included the axial compression ratio, the thickness of the wall (interface contact area), and the addition of a vertical grouted sleeve connection at the horizontal joint. Analyses and comparisons were conducted, in terms of the cracking propagation pattern, failure modes, force–displacement hysteretic curves, skeleton curves, bearing capacity, ductility, and plastic energy dissipation of PC walls. In addition, the prediction method for calculating the shear capacity of the PC wall with alveolar-type joints was presented based on the experimental results, and the calculated results are in good agreement with the experimental results.

## 2. Experimental Preparation

### 2.1. Specimen Preparation

Six precast concrete (PC) wall specimens with alveolar-type horizontal joints were designed and constructed, and they were labeled as PW1–PW6, respectively. The pseudo-static loading tests on these specimens were conducted contemporarily. Each specimen consisted of an upper concrete wall, a lower concrete wall, an alveolar-type horizontal joint, as well as a bottom beam, and the joint was arranged at 420 mm above the wall foot, which was similar to the PC wall panel structure proposed by [6,7,9,46]. In this case, for PW2–PW6 specimens, the vertical grouted sleeve was adopted at the horizontal joint to enhance the vertical connection behaviors of specimen. As a result, two D18 (18 mm in diameter) spliced steel bars were protruded from the lower wall and anchored into the grouted sleeves, which were embedded in the upper wall to assemble integral members. Its connection strength was approximately of 40% based on the principle of equal strength connection [24,47] so that the area of the spliced rebars is equal to the area of longitudinal bars inside the wall. A relatively weaker connection behavior at the joint is evident, compared to that of an equal strength connection for a structure [24] which meets the performance-based design methodology [7] and exhibits good seismic behaviors [6]. The overall height of all specimens was 1140 mm with a concrete cover of 20 mm, the length of wall was 600 mm, and the section dimensions of bottom beam was 300 mm × 400 mm. There was no vertical spliced rebars anchored into grouted sleeves at the horizontal joint of PW1 to investigate the differences between the specimen with or without grouted sleeve connection. Moreover, the interface contact area at the joint and the axial compression ratio are crucial influencing factors on the seismic behavior for the multi-story PC wall panel structural system [2,45,46]. To this end, the thickness of PW1–PW4 were 200 mm, whereas the thickness of PW5 and PW6 were chosen as 160 mm and 120 mm, respectively, to investigate the impacts of the interface contact area. Meanwhile, the axial compression ratio was chosen as 0, 0.1, and 0.2, respectively, based on precast concrete walls on the top, middle, and bottom floors of the multi-story wall panel structure in practical engineering, and the shear span ratio λ was 1.73, which was consistent with the requirements of Chinese code [47]. The main parameters of PW1–PW6 are summarized in Table 1. All specimens were cast of the same C30 grade concrete, and the HRB400 rebars were performed for longitudinal bars and horizontal distribution reinforcements, as well as the spliced rebars anchored into grouted sleeves. For the PW1 specimen, after the upper and lower walls were in place, they were assembled into the integral PC wall using the cement–mortar pressure grouting method at the horizontal joint. Meanwhile, for PW2–PW6, the cement–mortar bedding layer of 20 mm was firstly pressed at the joint by relying on self-weight of the upper wall, and then the high-strength grouting materials with non-shrinkage were grouted into the sleeves to realize the flexible connections between the upper and lower walls and form integrity. As noted previously, the alveolar-type joint, which was high at both ends and low at the middle, was adopted at the horizontal joint, and the specific geometric sizes and layout of reinforcing bars for each specimen are shown in Figure 2.

As described previously, for the precast walls with an alveolar-type horizontal joint, which were connected by tightening screw before the upper wall cast, the bottom rebars in the lower wall were inserted into the grouted sleeves, and then the self-leveling grouting material was fully injected into the sleeves after the upper concrete wall and lower wall were assembled. Besides, the bottom surface and the top surface of PC walls were roughened to an amplitude of about 6 mm during the manufacturing process of concrete members in order to enhance the friction or roughness at the interface, based on the requirements of Chinese standard “Technical Specification for Precast Concrete Structures (JGJ 1-2014)” [47].

### 2.2. Material Mechanics Properties

The designed strength grade of concrete was C30, and its compressive strength (standard 150 mm cube) was 36.9 MPa after 28-day curing. Twelve HRB400 D12 (12 mm in diameter) deformed bars were used as the longitudinal reinforcing bars, and the HRB400 D8 deformed bars, with spacing of 100 mm, were used as stirrups or horizontal distribution reinforcements. Meanwhile, two HRB400 D18 deformed bars were adopted as spliced rebars at the horizontal joint. The mechanical properties of reinforcement are listed in Table 2.

In addition, the GT/CT series grouted sleeve was adopted in this test to realize the flexible connection for PW2–PW6 at horizontal joint. It meets the specific requirements [14] in terms of the mechanical properties, including the tensile strength (exceeds 600 MPa) and elongation (no less than 16%) of sleeves. The compressive strength of the high-strength non-shrinkage grouting material was 89.2 MPa after 28-day curing. The standard cube of mortar was cast based on the specification requirements of JGJ/T70-2009 [48], and the compressive and tensile strengths measured after 28 days were 38.2 MPa and 2.6 MPa, respectively. Meanwhile, the elastic modulus and Poisson’s ratio of mortar were 2.52 × 10^4^ MPa and 0.18, respectively.

### 2.3. Test Setup and Measure Point Layout

To actually simulate the boundary and deformation conditions of the possible specimens, the multi-functional array loading test and analysis system were adopted to the pseudo-static test, and the schematic diagram of loading for each specimen, as shown in Figure 3a. The lateral load and constant vertical loading were, respectively, applied at the lateral side and at the top of wall, in accordance with the requirements of Chinese code JGJ/T 101-2015 [49]. Each specimen was anchored through the bottom beam, and the lateral direction was loaded by a 100 T MTS electro-hydraulic servo actuator. A photograph of the loading is shown in Figure 3b. For this purpose, the vertical load was firstly applied to the predetermined value and then kept constant by monitoring the force sensor on the jack, and the axial compression ratio was controlled to be 0, 0.1 and 0.2 respectively for PW1–PW6, based on the JGJ/T 101-2015 specification [49]. The lateral loading point was placed 1.04 m away from the wall foot, and the vertical load acted on the top of the wall. To this end, the force–displacement hybrid loading mechanism was adopted for specimens during the test [49], as shown in Figure 4. In this case, force control was adopted before the specimen was yielded, the increments of force load at each stage were 5 kN or 10 kN, and the load was cycled once at each stage. Nevertheless, the displacement load control was conducted after the specimen was yielded, the increments for the displacement load of each stage reached a yield displacement Δ_y_, and the load was cycled twice at each stage. Once the specimens were damaged and the lateral load dropped to 85% of the peak load, the test stopped. In addition, the forward direction of the actuator was positive and the backward direction of the actuator was negative during the test.

The layouts of the PW1–PW6 measure points, composed of rebars strain gauges and displacement meters (LVDTs), were basically identical, and the detailed location of the measure points is plotted in Figure 5. A total of 12 reinforcement strain gauges attached to the outer edge of longitudinal bars were located 20 mm above and below the horizontal joint, and 20 mm above the bottom beam, and were labeled as S1 to S12, respectively. Moreover, for the PW2–PW6 specimens, four rebars strain gauges (marked as T1~T4) were pasted at the splied rebars located at 20 mm of the upper and lower grouted sleeves, respectively, to demonstrate the connection reliability between the rebars and the grouted sleeves, as shown in Figure 5a,b. Meanwhile, the lateral displacement of the loading point was recorded directly by the actuator. Figure 5c clearly shows that six LVDTs were installed on the right side of each specimen, labelled W1 to W6 from the top to the bottom, while W1–W3 were installed at the wall side from the loading point to the horizontal joint with a spacing of 275 mm. Moreover, W4 and W5 were employed to measure the lateral displacement 50 mm and 250 mm below the horizontal joint, respectively, and W6 was used to measure the horizontal slippage of the bottom beam.

## 3. Test Process and Failure Modes of Specimens

The mechanical properties and failure modes of the precast concrete (PC) wall panel specimen with an alveolar-type horizontal joint were predicted before the pseudo-static loading test. The specimen’s design and manufacture meet the demands of anchoring, and it was assumed that the spliced rebars anchored into grouted sleeves exhibit good anchorage properties; thus, during the test, there are no anchoring destructions and reinforcements cannot slide about. As a result, Figure 6 describes the force diagram when the constant vertical loading and the lateral loading were, respectively, applied at the top and lateral side of wall, as shown in Figure 6.

As seen in Figure 6, the horizontal load *P* was mostly balanced by these forces, including the oblique pressure force of the compressive zone concrete, and the dowelling function provided spliced rebars anchored into grouted sleeves, as well as the horizontal friction resistance at the horizontal joint, etc. Meanwhile, the compression zone concrete of the lower wall close to the horizontal joint was subjected to combined action of axial force and horizontal friction, which made it easier to find diagonal cracks, fractures, or even crushes, due to the principal tensile stress (Marked A in Figure 6). As a result, once the friction resistance (the function of both the area at the interface and axial force) is overcome and the upper wall starts to slip or rotate, lateral deformation of the wall is achieved and an overload at the wall corner is created, eventually causing the collapse of concrete wall, as similar to rocking wall’s behavior [50].

To this end, Figure 7 depicts the final failure mode of PW1–PW6. As clearly seen from Figure 7, the failure mode of each specimen is basically similar after loading; thus, Figure 8 only shows the evolution of cracks for PW4 during various loading cycles. All specimens underwent the elastic stage without cracking during the early loading stage, and the load increased linearly with displacement. Furthermore, the load–displacement relationship curves of loading or unloading were essentially coincident. As the load increased, the horizontal cracks began to occur at the horizontal joint at the tension zone, which then gradually expanded or even ran through along the joint. It was also observed that the horizontal cracks at the joint changed and diagonal cracks under the larger load were developed. It was indicated that specimen with a grouted sleeve connection can reach the yield stage once the longitudinal bars of the tensile side yielded. As the load continued to increase, the compressive zone concrete located at the lower wall was crushed or spalled off, and the longitudinal reinforcement below the horizontal joint was exposed. The destructions of the compressive zone concrete and, evidently, the diagonal cracks located at the lower wall were remarkable. In addition, the shear failure at the horizontal joint was observed for specimen in the ultimate load, as shown in Figure 8. In this case, for specimens with axial compression, the numerous oblique main cracks at surface of the lower wall were also observed, and meanwhile several visible diagonal cracks extended at the surface of upper wall. However, at first, the horizontal cracks occurred at the horizontal joint. Then, they developed rapidly and gradually changed to shear failure following the increase in the load for specimens without axial compression.

In the loading process, the lateral load of PW1–PW4 were 70 kN, 30 kN, 90 kN, and 120 kN, respectively, when the first crack of the specimen appeared. Meanwhile, the cracking load of PW5 and PW6 were 65 kN and 50 kN, respectively. This indicates that the grouted sleeve connections at the horizontal joint, the axial compression ratio, and thickness of the wall all have a significant effect on the cracking load. The cracking load increased significantly as the axial compression ratio or additional grouted sleeve connections were enhanced. This was because the enhancement of the axial compression ratio was equivalent to provide the vertical preload pressure to the PC wall, which greatly delayed the cracking and extension of the concrete at the tensile side of the joint under the same loading conditions. Therefore, the occurrence of horizontal slippage at the horizontal joint was delayed and the cracking load was dramatically enhanced. Furthermore, the spliced rebars anchored into grouted sleeves were employed to improve the bending deformation for the specimen, which reduced or eliminated the horizontal slippage along the joint. It was also found that the horizontal joint was easier to crack and destroy, and PW2 saw a significant slip, as shown in Figure 7b. It was observed that the number of diagonal cracks at the lower concrete wall significantly increased with the enhancement of the axial compression ratio, and these cracks rapidly developed and then gradually extended downward diagonally. The cracking mode was changed from horizontal cracks at the joint to diagonal cracks at the lower concrete wall for specimens with axial compression, and the angle of the diagonal cracks ranged from 33° to 42°. In addition, these angles increased gradually as the axial compression ratio and the grouted sleeve connections were added, as shown in Figure 7a,c,d. The reason is that the lateral stiffness and shear strength after the specimen cracked were strengthened by the increasing axial compression ratio. As a result, the formation and development of cracks, as well as the slippage at the horizontal joint, was delayed or decreased under the same loading, which significantly changes the extension direction of the crack and the mechanical properties of specimens. Meanwhile, the compressive zone concrete provided significant diagonal compression struct action for specimens with larger axial compression, and the dowelling function provided by the spliced rebars anchored into the grouted sleeves. This drastically improves the plastic deformation and bearing capacity of the specimen, as shown in Figure 7c,d. Moreover, the cracking load decreases gradually as the reduced thickness of the wall, as shown in Figure 7e,f.

As a result, the horizontal cracks appeared, developed along the horizontal joint in the loading process, and extended gradually in a diagonal direction after the axial compression ratio increased or the grouted sleeve connections were added, which caused numerous visibly diagonal cracks at the lower wall and the spalling or even collapse of concrete. This then drastically slows down the destruction at the horizontal joint, and the shear failure changed from the horizontal joint to the lower wall. Meanwhile, few or several visible diagonal cracks were also observed at the surface of the upper wall. In addition, for specimens with grouted sleeve connections, the visible oblique cracks and slightly horizontal cracks were also found at the surface of the lower concrete wall due to the enhancement of flexural strength, as shown in Figure 7a,c–f. The failure modes of PW1–PW6 were shear failure, and the cracking modes and the local failure characteristics at the horizontal joint were significantly affected by the axial compression ratio and vertical sleeve grouted connection, whereas the wall thickness only had a slight impact.

## 4. Results and Discussion

### 4.1. Hysteresis Curves

Figure 9 depicts the lateral force–displacement hysteresis curves of PW1–PW6, which were measured directly through the force and displacement sensors in the actuator at top of the wall. It was observed that the relationship curves between the lateral force and the displacement were narrow and, before the specimen cracked, were linear. As the load repeats and further increases, the area enclosed by hysteretic loops and the residual deformation of the specimen began to occur and increase continuously, and the secant stiffness, as well as the slope of load–displacement curves, which descended gradually after the specimen yielded. To be noted, the degeneration rate of displacement is larger than that of the load as the loading and repeating times increased after the specimen yielded, and the plastic deformation started to gradually increase as the displacement load increased. Meanwhile, the stiffness and bearing capacity of the second cycle were less than those of the first cycle under the same displacement load stage, indicating that the degeneration rate of stiffness increased significantly and that the energy dissipation capacity dropped gradually following an increase in the cycle times. These results were also consistent with those of [51].

Figure 9 clearly shows that the plumpness of hysteretic curves for PW1 were much less than those of PW3, which means that the ductility, energy dissipation ability, and bearing capacity can be significantly enhanced by grouted sleeve connections at the horizontal joint, as shown in Figure 9a. As a result, the grouted sleeve connection increased the interface contact area of the compressive zone concrete at the horizontal joint; thus, the shear stress of the interface was descended. The spliced rebars anchored into grouted sleeves provided the flexural stiffness and dowelling function for specimens subjected to lateral displacement or rotation, which deforms the wall and further decreases or eliminates the slippage along the horizontal joint. Therefore, the brittle shear failure along the horizontal joint was delayed evidently, which then improves the energy dissipation capacity and ductility, as mentioned in [7,52]. Furthermore, it was also found that the hysteretic loops of PW2 were relatively flat and less than those of other specimens, thus revealing the pinching effect phenomenon. However, the hysteretic curves of PW3 and PW4 were relatively plump, smooth, and full shuttle-shaped, and the areas enclosed by them were much greater than those of PW2, as shown in Figure 9b,d. Therefore, it was indicated that the axial compression ratio has a significant effect on the energy dissipation, and the specimen with axial compression has good plastic deformation and energy dissipation capacity, as well as larger bearing capacity. This is because the axial compression ratio can significantly delay the horizontal slip failure and enhance the lateral stiffness after the specimen cracked; therefore, the plastic deformation capacity, bearing capacity, and energy dissipation ability are drastically improved.

In addition, the areas enclosed by the hysteresis loops of PW5 and PW6 were less than those of PW4; thus, as wall thickness reduced, the energy dissipation capacity gradually decreased and the bearing capacity also significantly declined, as shown in Figure 9e,f. For this purpose, the interface contact area at the horizontal joint descended drastically due to a reduction in the wall thickness, which caused the shear stress of the interface to continuously increase and the lateral stiffness to significantly decline. As a result, the vertical grouted sleeve connection, axial compression ratio, and thickness of the wall all have significant impacts on the hysteretic curves and energy dissipation ability after the specimen yielded.

### 4.2. Skeleton Curves and Bearing Capacity

The experimental results of each specimen are listed in Table 3. The peak load is the maximum lateral load during the loading process, and the ultimate load is the corresponding load when the lateral load drops to 85% of the peak load or the specimen becomes damaged. The yield displacement Δ_y_ is calculated by the energy equivalence method [51]. The area enclosed by the bilinear BDE then equals the area enclosed by actual *P*-Δ curve OAB, as shown in Figure 10. The yield displacement is determined by the formula Δ_y_ = 2 (Δ_max_ − *A*/*P*_max_), where *A* is the area enclosed by the curve OBE, and *P*_max_ and Δ_max_ are the corresponding peak load and peak displacement, respectively.

Figure 11a and Figure 11b, respectively, depict the skeleton curves of PW1–PW4, PW3, PW5, and PW6. As seen in Figure 11, the stress–strain process of each specimen underwent the elastic stage, as well as the elastic–plastic and destruction stages. The slope of skeleton curves were almost constant and coincident when the force load was less than 30 kN, indicating the relationship between the force and linear displacement for the specimen during the elastic stage. Accordingly, the vertical grouted sleeve connection at the horizontal joint, the axial compression ratio, and the thickness of the concrete wall have no significant influence on the skeleton curves during the elastic stage. However, when the force load exceeded 30 kN, differences between the skeleton curve of PW1–PW6 were observed and then gradually increased. After the specimen yielded, the bearing capacity and ductility were enhanced rapidly by increasing the axial compression ratio or by adding vertical grouted sleeve connections at the horizontal joint, as shown in Figure 11a. Due to the addition of vertical grouted sleeve connections or the enhancement of the axial compression ratio, the shear stress of the interface was decreased, which caused the flexural strength or lateral stiffness to significantly increase; therefore, the brittle shear failure at the horizontal joint was evidently delayed. In addition, the dowelling function was provided by spliced rebars anchored into grouted sleeves at the horizontal joint due to slippage at the interface, which also greatly enhanced the bearing capacity and ductility of the specimens, as mentioned in [7].

Figure 11b also plots the skeleton curves of PW3, PW5, and PW6 to further investigate the effect of wall thickness (interface contact area) on the skeleton curves. It can be obviously seen that the bearing capacity of the specimen significantly decreases as the wall thickness decreases. This is due to the significantly reduced interface contact area of the horizontal joint and the rapidly increased shear stress of the concrete interface following a reduction in the wall thickness, which caused the concrete located at the corner of the lower wall and the surrounding interface to crush or spall off for the specimen under same lateral load, which also thereby reduced the bearing capacity of the specimens.

In addition, as seen in Table 3, compared with PW1, the cracking load, yield load, and peak load of PW3 were enhanced significantly, increased by 28.6%, 50.6%, and 41.9%, respectively. As such, the bearing capacity obviously increased when vertical grouted sleeve connections were added at the horizontal joint, and the maximum increased by 41.9%. Meanwhile, a comparison of the skeleton curves of PW2~PW4 revealed that the cracking load, yield load, and peak load of the specimens also increased significantly following the enhancement of the axial compression ratio. Accordingly, compared with the specimen without axial compression, the bearing capacity of specimens could be drastically enhanced due to the increase in the axial compression ratio, which increased by 2.2 times. It was also noticed that the bearing capacity of the specimen gradually decreases with a reduction in the wall thickness (interface contact area), and the maximum decreased by 35.7%.

### 4.3. Prediction Method for the Shear Capacity of PC Walls

Several basic assumptions are proposed to calculate the shear capacity of precast concrete (PC) walls with horizontal alveolar-type joint, based on some relevant investigations which employed the predication method [6,45,46]. (a) These contributions, including the diagonal compressive zone concrete wall, friction resistance at the horizontal joint, and spliced rebars, anchored into grouted sleeves together to provide the shear bearing capacity of PC walls. (b) The failure mode of the PC walls revealed that the inclined compressive zone concrete of walls was crushed and caused shear failure at the horizontal joint. (c) The horizontal distribution reinforcements had not yielded after the test, and the failure of the interface occurred before the overall failure of the wall. The effects of the horizontal distribution reinforcements on the shear capacity of specimens should be ignored. Three basic assumptions are almost consistent with the failure modes and experimental results of PC walls with horizontal alveolar-type joints. As described previously, the diagonal compressive zone concrete provides the main shear capacity for PC walls with axial compression, as shown in Figure 12. The shear capacity, also provided by dowelling functions of spliced rebars, anchored into grouted sleeves at horizontal joints and the interface friction resistance after slippage between the upper and lower walls. Therefore, it can be finally summarized that, during the loading, the shear capacity of the PC wall was mainly provided by forces linked to the diagonal compression strut actions, dowelling functions of spliced rebars, and friction resistance at the interface.

The compressive stress of the concrete in the diagonal compression strut was determined as follow:(1)$$\sigma_{s}=\gamma f_{c}$$

The strength reduction coefficient formula [45], taking into account the effect of the concrete cracking process on diagonal compression strut, can also be performed here.
*γ* = 0.7 − *f*_c_/200(2)
where *f*_c_ presents the axial compressive strength of concrete, and *f*_c_ = 14.3 N/mm^2^ represents concrete designed with a strength grade of C30. As a result, the diagonal compression strut actions was derived from horizontal balanced equation [46]:(3)$$P_{c1}=\sigma_{s}b(h_{0}-H\;\cot\theta )\cos\theta \sin\theta$$
where *b*, *h*_0_, and *H*, respectively, represent the thickness, the effective width of the wall section, and the height of wall. *θ* represents the angle of diagonal cracks, and the angle of the actual inclined cracks ranges from 33° to 42° in the test. The shear-span ratio calculated by $\lambda =\frac{M}{Ph_{0}}=\frac{PH}{Ph_{0}}=\frac{H}{h_{0}}$, when *λ* exceeds 0.5, and the value of *λ* is directly taken as 0.5, based on the values specified in [45]. Therefore, Equation (3) was simplified to:(4)$$P_{c1}=\frac{1}{2}\sigma_{s}bh_{0}\sin2\theta (1-\lambda \cot\theta )$$

The interface friction resistance is greater than the horizontal load for specimens with larger axial compression; thus, the specimen will not suffer sliding destruction due to the insufficient friction force at the horizontal joint. However, for specimens without axial compression, the cracks of horizontal joint will appear immediately and will develop and change rapidly to brittle shear failure before the overall failure of the specimen. Furthermore, the cohesive force at the interface provides shear capacity before the horizontal joint crack. Therefore, the interface friction P_c2_ and the cohesive force P_c3_ were determined, respectively, by the following Equations (5) and (6).
(5)$$P_{c2}=\mu N_{m}$$
(6)$$P_{c3}=\alpha_{c}f_{c}A_{c}$$
where the friction coefficient *μ* in the test is taken as 0.8 for this interface, after referring to the standards of ACI318-19 [30] and experimental results. *N*_m_ represents the vertical pressure at the interface. *α*_c_ represents the combination coefficient of the stick-locking effect, while *α*_c_ = 0.025 represents the alveolar-type joint, based on the results summarized in [53]. *A*_c_ represents the effective area at the contact interface of the concrete wall, *A*_c_ = *bh*_0_ = 1.2 × 10^5^ mm^2^.

However, the cohesive force no longer contributes to the lateral resistance for shear bearing after the slippage at the interface, though the interface friction will continue to provide the shear capacity. In addition, the spliced rebars anchored into grouted sleeve immediately provide the dowelling functions once the slippage occurs at the horizontal joint. The calculated formula for the shear capacity provided by spliced rebars was proposed as Equation (7), based on the literature [47].
(7)$$P_{c4}=\frac{0.5}{{\left(\lambda +1\right)}^{2}}(f_{y}A_{s}+0.5f_{y}{}^{\prime }A_{s}{}^{\prime })$$
where *λ* represents the shear span ratio, which is directly taken as 0.5 for *λ* = 1.73 in this test. *f*_y_ and *f*_y_*^′^*, respectively, represent the yield strength of the tensile and compressive design for spliced rebars, i.e., *f*_y_ = *f*_y_^′^ = 360 MPa for HRB400 rebars [47]. *A*_s_ and *A*_s_^′^ are the corresponding section areas of rebars, i.e., *A*_s_ = π (18)^2^/4 mm^2^. As a result, the Equation (7) is simplified to:*P*_c4_ = 0.1 *n*_s_*f*_y_*A*_s_(8)
where *n*_s_ represents the number of spliced rebars through the horizontal joint.

As a consequence, the formula for calculating the shear capacity of precast walls was derived from Equation (1) to Equation (8).
(9)$$P_{\mathrm{am}}=\frac{1}{2}\gamma_{s}\sigma_{s}bh_{0}\sin2\theta (1-\lambda \cot\theta )+\mu N_{m}+0.1n_{s}f_{y}A_{s}$$
where *γ*_s_ represents the effective width coefficient of the diagonal compression strut at compressive zone concrete, and *γ*_s_ = 0 when *P* > *μN*_m_. However, when *P* < *μN*_m_, n_s_ is directly taken as 0.

The shear capacity of all PC walls is calculated by the Equation (9), and these results are summarized in Table 3. As a result, it can be finally concluded that the calculated results are in good agreement with the experimental results.

### 4.4. Ductility Factor

Figure 13 plots the variable curves of the ductility factor for all specimens, based on the experimental results in Table 3. Figure 13a shows that adding vertical grouted sleeve connection at the horizontal joint can significantly improve the ductility factor, which increased by 19.4%. Meanwhile, the ductility factor of specimens with an axial compression ratio of 0.2 was still much greater than that of specimens without axial compression, indicating that the ductility factor could be enhanced by increasing the axial compression ratio by 73.2%. It was also observed that the ductility factor increases first and then decreases following the increase in the axial compression ratio, similar to [2]. The increase in axial compression ratio enhanced the vertical pre-pressure and lateral stiffness, delayed horizontal cracking or slippage, and changed the cracking mode; thus, the ductility and bearing capacity of the specimen increased drastically. However, as the axial compression ratio increased further, the cracks immediately closed due to the larger axial compression ratio when the specimen after cracking was subjected to the same lateral loading conditions. In this case, the contribution of rebars to energy dissipation slightly decline. Meanwhile, concrete mainly acts as bearing material to resist the bending moment and the shear force. Accordingly, the principal compressive stress is relatively larger under the same displacement load for specimens with a larger axial compression of 0.2, making it easier to reach the failure displacement than that of specimens with an axial compression of 0.1. Moreover, as seen in Figure 13b, it is also found that the ductility factor of the specimens barely changed as the thickness of the concrete wall decreased. As a result, it is concluded that the thickness of the wall has no significant effect on the ductility factor.

In addition, to demonstrate the reliability and consistency of connection properties for spliced rebars anchored into grouted sleeves at horizontal joint, Figure 14 describes the force–strain hysteretic curves at the upper and lower rebars connected by grouted sleeves in PW3, which were then marked as T1 to T4 strain measure points (Figure 5).

As seen from Figure 14, the force–strain curves of the upper and lower rebars connected by grouted sleeves are relatively close and flat during the elastic stage. As the load repeats and increases, the strains of upper and lower reinforcements increase significantly and the differences between of them increase rapidly. Furthermore, the lower rebars’ strains increased drastically more than those of upper rebars. In this case, the lower rebars’ strains are much larger than those of upper rebars for specimens in ultimate state and the strain value of lower rebars exceeds 5500 με. However, the strain value only reaches 2400 με for upper rebars, and the results are similar to those in [14,15]. Therefore, the spliced rebars connected by grouted sleeves can effectively transfer the longitudinal rebars’ stress during the test.

### 4.5. Energy Dissipating Capacity

The area *E* of hysteretic loops and the equivalent viscous damping coefficient *h*_e_ are commonly used to evaluate the energy dissipation capacity of structures [6,26]. Figure 15a and Figure 15b, respectively, depict the relationship curves of the accumulated energy dissipation and displacement load for PW1–PW6, and the equivalent viscous damping coefficient of all specimens in the ultimate stage. As seen from Figure 15a, the energy dissipation gradually increased as the displacement load increased, and particularly after the specimen yielded. However, the accumulated energy dissipation was relatively slight and flat before the specimen yielded. Accordingly, it was indicated that the axial compression ratio, the vertical grouted sleeve connections, and the thickness of wall had no significant effects on the energy dissipation capacity for the specimen pre-yield, whereas those influencing factors were significant after the specimen yielded. It was also found that the energy dissipation of the specimen with a grouted sleeve connection was obviously greater than that of the specimen without a grouted sleeve connection in the ultimate state. As a result, the accumulated energy dissipation of specimen can be drastically enhanced by adding the vertical grouted sleeve connection, which increased by 164.2%. Meanwhile, it can also be seen from Figure 15a that the energy dissipation obviously increased as the axial compression ratio continuously increased by 783.1%. Moreover, the accumulated energy dissipation of the specimen significantly reduced after wall thickness, decreased by 62.9%.

In addition, as seen in Figure 15b, the equivalent viscous damping coefficient for specimens in the ultimate state significantly increased following the 64.3% increase in the axial compression ratio. Similarly, the equivalent viscous damping coefficient was drastically improved by adding a grouted sleeve connection at the horizontal joint, and the maximum increased by 32.4%. Meanwhile, the equivalent viscous damping coefficient gradually declined as the thickness of the concrete wall, reduced by 32.6%. Therefore, increasing the axial compression ratio and adding grouted sleeve connection at the horizontal joint can observably enhance the equivalent viscous damping coefficient of the specimen. This is due to the formation and development of cracks in the tensile side of concrete under the same load, which was delayed by an increase in the axial compression ratio and the addition grouted sleeve connection, which slowed down the horizontal slippage of the interface and changed the cracking mode of specimens. Moreover, the compressive zone concrete provided significant diagonal compression struct actions for specimens with a larger axial compression ratio, and the dowelling function was provided by spliced rebars anchored into grouted sleeves at the joint after the slippage between the upper and lower walls, which drastically improves the ductility factor, the bearing capacity, and the energy dissipation capacity of specimen.

## 5. Conclusions

In this work, six precast concrete (PC) wall specimens with an alveolar-type horizontal joint were designed and constructed, and pseudo-static tests on these specimens were conducted to investigate the impacts of influencing factors, such as the axial compression ratio, the thickness of wall (interface-contact area), and the addition of vertical grouted sleeve connection at the horizontal joint, on the seismic behaviors of PC walls. These seismic performance indicators, including failure modes, load-bearing capacity, ductility factors, and energy dissipation capacity of PC walls, were compared and analyzed, and the main conclusions are drawn as follows.

For the failure modes of the PC walls with an alveolar-type horizontal joint, the compressive zone concrete located at the lower walls was crushed or spalled off, and the longitudinal reinforcements below the horizontal joint were exposed to buckling. Meanwhile, the destructions of the compressive zone concrete and diagonal cracks at the lower walls were significant. Furthermore, the shear failure at the horizontal joint was observed for PC walls under the ultimate load, which was different from the failure modes of the cast-in-place concrete wall.The axial compression ratio and the vertical grouted sleeve connection at the horizontal joint had a significant influence on the cracking mode of PC walls, whereas the effect of the wall thickness was only small. For specimens with axial compression, oblique cracks at the surface of the lower wall were observed, and several visible diagonal cracks extended at the surface of upper wall. However, these horizontal cracks occurred at the joint first and then developed and changed rapidly to brittle shear failure following the increase in the load for specimens without axial compression. The shear failure at the horizontal joint drastically delayed and improved following an increase in the axial compression ratio and the addition of a grouted sleeve connection at the horizontal joint.Numerous visible diagonal cracks and slight horizontal cracks at the lower wall were observed for specimens with vertical grouted sleeves connection. The cracking modes changed from horizontal cracks at the joint to inclined cracks at the lower wall following the enhancement of the axial compression ratio. The specimen exhibits better ductility after adding the vertical grouted sleeve connection at the horizontal joint, as it increased by 19.4%.The axial compression ratio, vertical grouted sleeve connection, and thickness of the concrete wall are also crucial factors which affect the seismic behaviors of PC walls. The bearing capacity and energy dissipation capacity are significantly improved by increasing the axial compression ratio or adding vertical grouted sleeve connection, which increased by 41.9% and 64.3%, respectively. However, the ductility factor firstly increases and then decreases following the increase in the axial compression ratio, as the maximum increased by 73.2%. The reduction in the wall thickness has significant effects on the shear capacity and energy dissipation capacity of the PC walls, which decreased by 35.7% and 62.9%, respectively, whereas the effect on ductility was only small.The prediction method for calculating the shear capacity of precast concrete walls with alveolar-type horizontal joint was proposed based on the experimental data. Furthermore, these calculated results are in good agreement with the experimental results.

## Figures and Tables

**Figure 1 materials-15-02301-f001:**
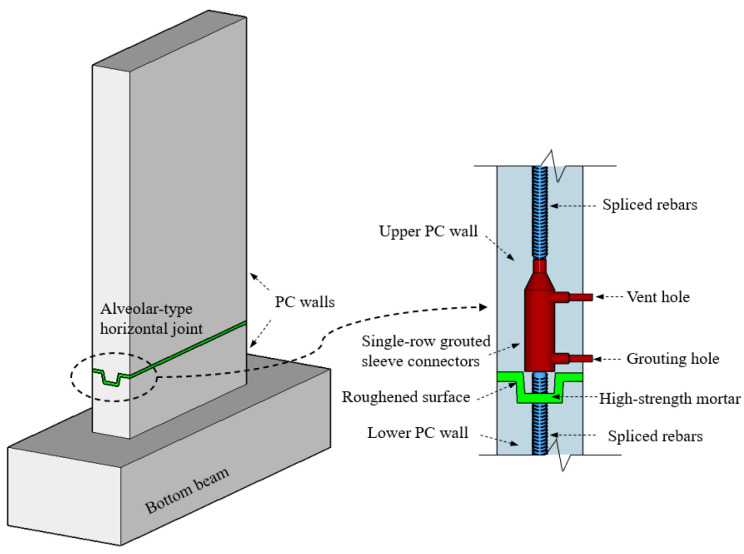
Schematic diagram of the horizontal alveolar-type joint.

**Figure 2 materials-15-02301-f002:**
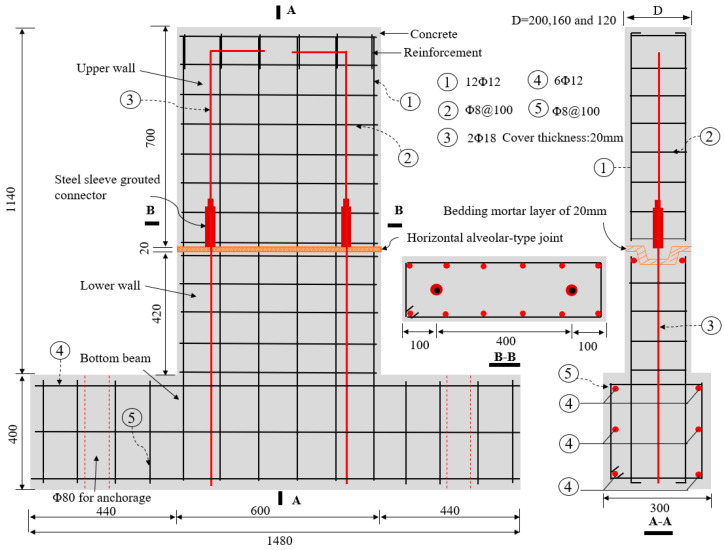
Geometric sizes and layout of reinforcing bars for specimens (in mm).

**Figure 3 materials-15-02301-f003:**
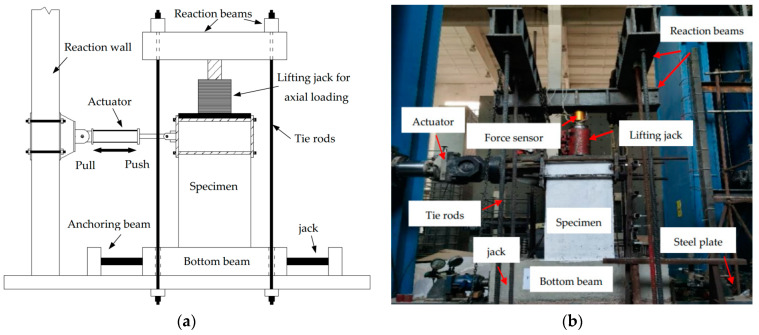
Schematic diagram of the loading device for specimens: (**a**) schematic diagram of the loading process and (**b**) a photograph of the setup.

**Figure 4 materials-15-02301-f004:**
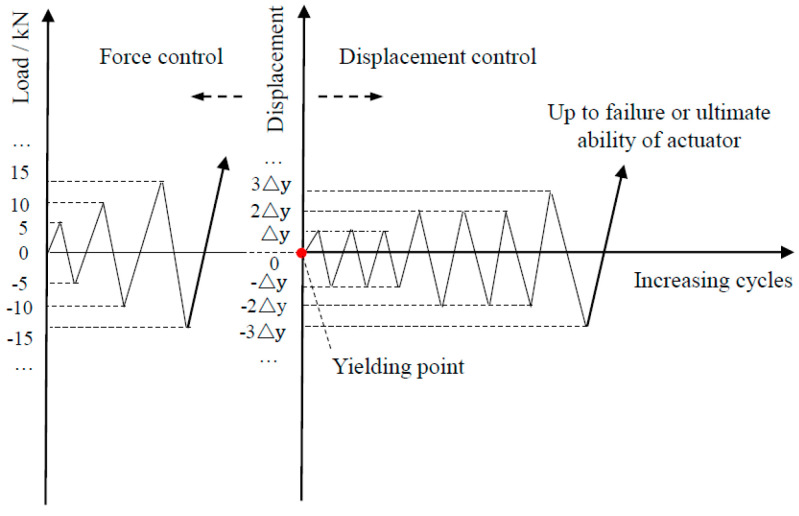
Lateral force-displacement history.

**Figure 5 materials-15-02301-f005:**
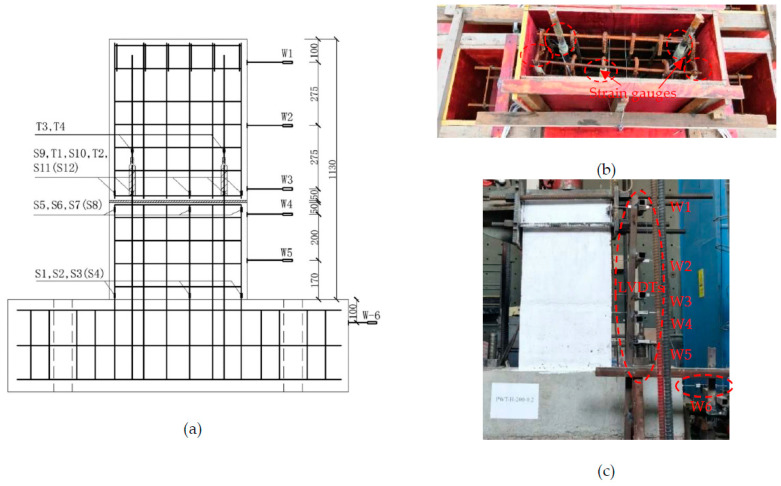
Instrumentation layout of specimens (in mm): (**a**) schematic diagram of the measure points; (**b**) photograph of rebar strains; and (**c**) displacement meter (LVDTs) layout.

**Figure 6 materials-15-02301-f006:**
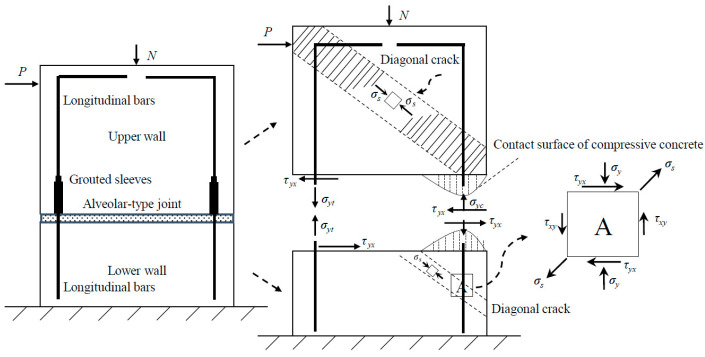
The force diagram of the PC wall.

**Figure 7 materials-15-02301-f007:**
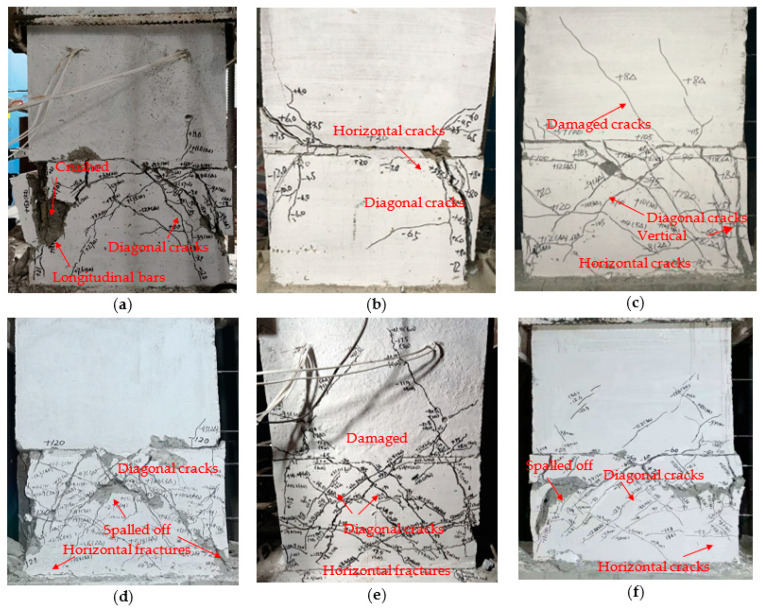
Failure modes of PW1–PW6: (**a**) PW1; (**b**) PW2; (**c**) PW3; (**d**) PW4; (**e**) PW5; and (**f**) PW6.

**Figure 8 materials-15-02301-f008:**
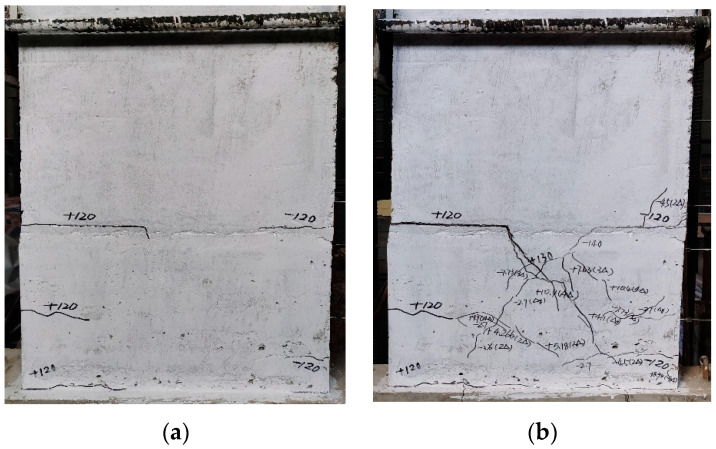
The evolution of the cracks for PW4 during the various loading cycles: (**a**) crack load; (**b**) yield load; (**c**) peak load; and (**d**) ultimate load.

**Figure 9 materials-15-02301-f009:**
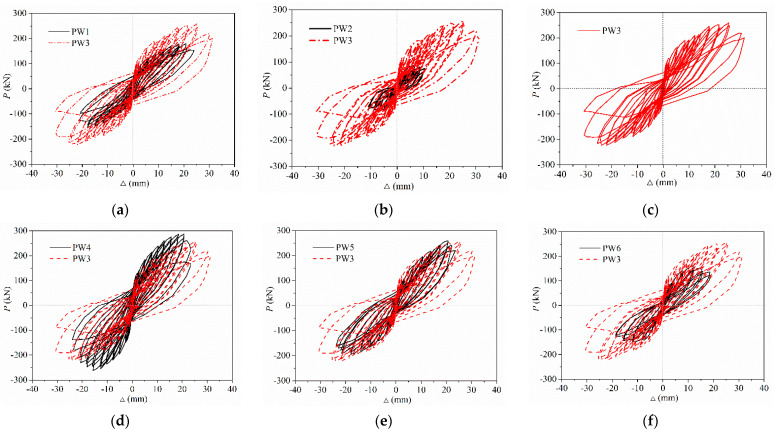
Hysteretic curves of PW1–PW6: (**a**) PW1; (**b**) PW2; (**c**) PW3; (**d**) PW4; (**e**) PW5; and (**f**) PW6.

**Figure 10 materials-15-02301-f010:**
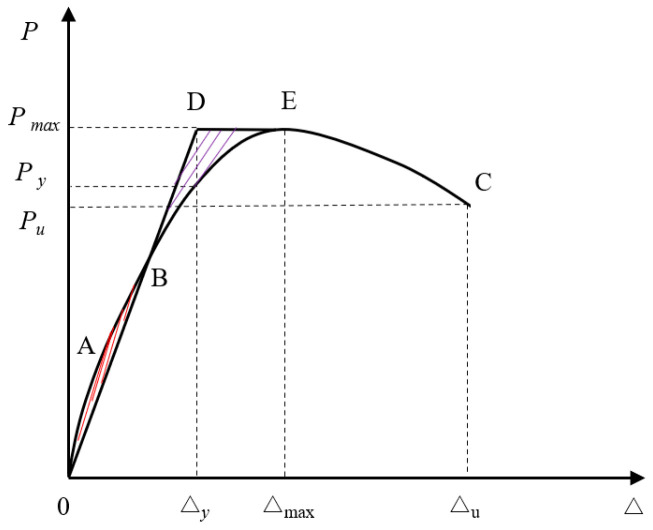
Energy equivalence method.

**Figure 11 materials-15-02301-f011:**
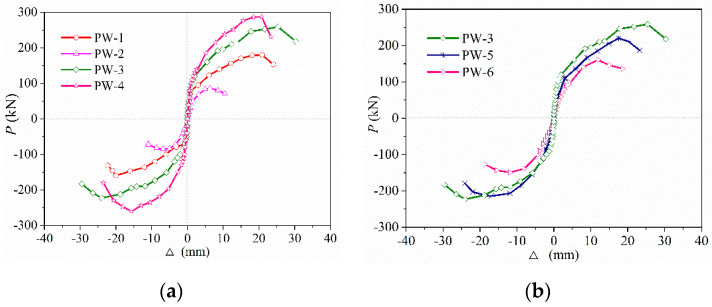
Skeleton curves of PW1–PW6: (**a**) PW1–PW4 specimens; and (**b**) PW3 and PW5–PW6 specimens.

**Figure 12 materials-15-02301-f012:**
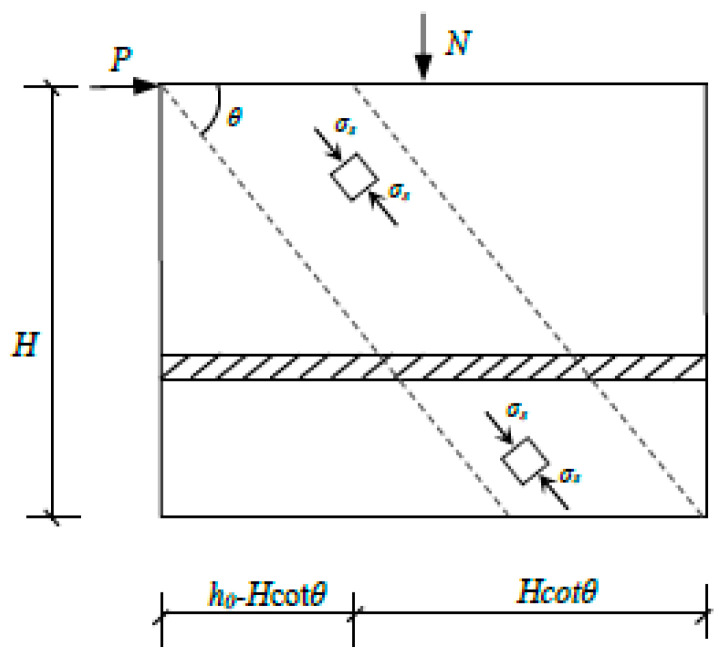
Diagram of diagonal compression strut.

**Figure 13 materials-15-02301-f013:**
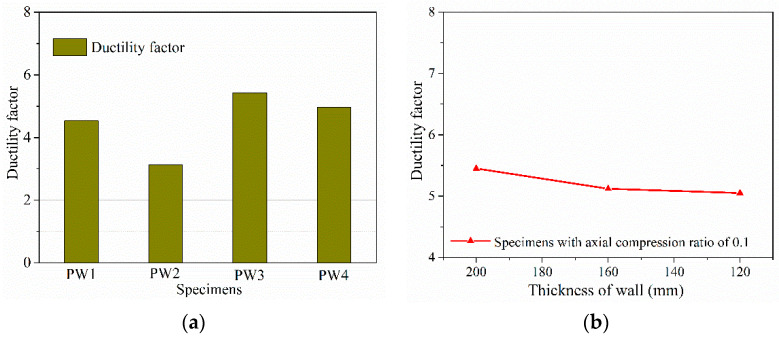
Variable curves of ductility factor for PW1–PW6: (**a**) ductility factor of PW1–PW4 and (**b**) relationship curves between the ductility factor and the concrete wall thickness.

**Figure 14 materials-15-02301-f014:**
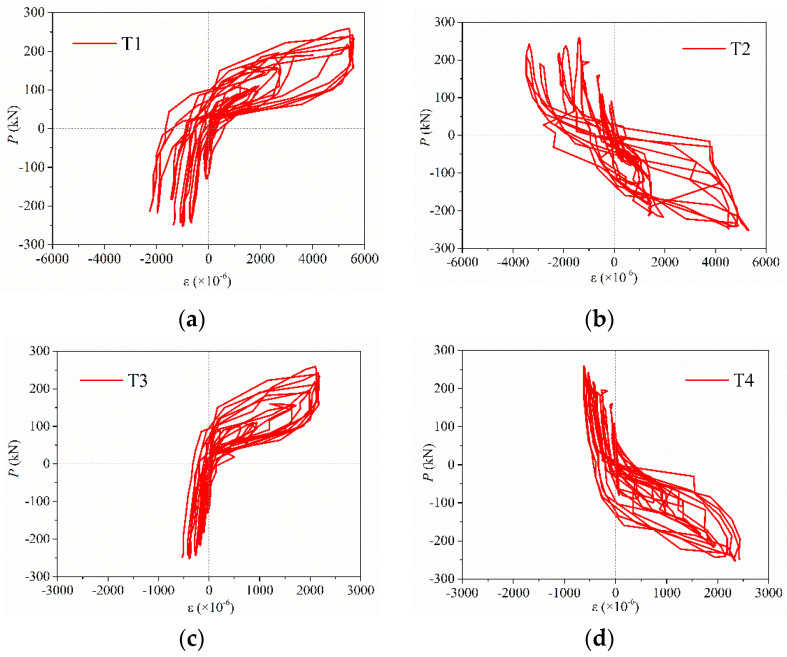
Force-strain hysteretic curves of T1–T4 for PW3: (**a**) T1 measure point; (**b**) T2 measure point; (**c**) T3 measure point; and (**d**) T4 measure point.

**Figure 15 materials-15-02301-f015:**
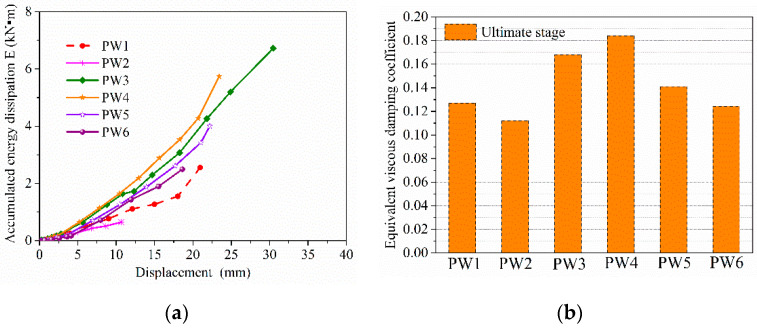
Energy dissipation and equivalent viscous damping coefficient of PW1–PW6: (**a**) relationship curves between the accumulated energy dissipation and displacement load and (**b**) the equivalent viscous damping coefficient for PW1–PW6 in the ultimate state.

**Table 1 materials-15-02301-t001:** Main parameters of PW1–PW6.

Mark	Shear Span Ratio	Axial Compression Ratio	Vertical Loading	Thickness of Wall (mm)	Rebars Anchored into Grouted Sleeves	Reinforcements of Wall	
			(kN)			Longitudinal Bars	Horizontal Reinforcements
PW 1	1.73	0.1	171.6	200	—	12Φ12	Φ8@100
PW 2	1.73	0	0	200	2Φ18	12Φ12	Φ8@100
PW 3	1.73	0.1	171.6	200	2Φ18	12Φ12	Φ8@100
PW 4	1.73	0.2	343.2	200	2Φ18	12Φ12	Φ8@100
PW 5	1.73	0.1	137.3	160	2Φ18	12Φ12	Φ8@100
PW 6	1.73	0.1	102.9	120	2Φ18	12Φ12	Φ8@100

**Table 2 materials-15-02301-t002:** Mechanical properties of rebars.

Steel Type	Diameter	Yield Strength	Tensile Strength	Elongation	Elastic Modulus
	(mm)	(MPa)	(MPa)	(%)	(MPa)
HRB400	8	436.2	556.5	19.2	1.98 × 10^5^
	12	458.6	577.1	22.1	2.03 × 10^5^
	18	466.3	574.9	24.2	2.11 × 10^5^

**Table 3 materials-15-02301-t003:** Test results of PW1–PW6.

Mark	Loading Direction	Crack Load		Yield Load		Peak Load		Ultimate Load		$\overset{–}{µ}$	*P*_am_ (kN)	*P*_m_/*P*_am_
		*P*_cr_ (kN)	Δ_cr_ (mm)	*P*_y_ (kN)	Δ_y_	*P*_m_ (kN)	Δ_m_	*P*_u_ (kN)	Δ_u_ (mm)			
					(mm)		(mm)					
PW1	Positive	70	0.75	110.08	4.98	179.0	19.97	152.1	24.03	4.54	158.5	1.07
	Negative	70	0.71	94.17	5.24	160.1	20.43	130.9	22.28			
PW2	Positive	30	0.77	71.92	3.48	86.2	6.74	73.3	11.47	3.13	70.9	1.21
	Negative	30	0.85	72.95	3.35	85.1	6.80	71.4	11.84			
PW3	Positive	90	0.92	162.5	5.75	258.3	25.34	219.6	30.3	5.42	266.2	0.91
	Negative	90	1.33	145.1	5.34	223.0	23.97	189.6	29.55			
PW4	Positive	120	2.65	179.6	4.85	287.5	19.51	243.1	23.3	4.96	302.6	0.91
	Negative	120	2.29	185.6	4.61	260.9	18.92	221.8	23.5			
PW5	Positive	65	0.98	130.7	5.29	220.2	17.56	187.1	23.21	5.12	230.3	0.94
	Negative	65	0.94	127.9	4.12	214.4	16.95	182.2	24.08			
PW6	Positive	50	1.36	97.3	3.82	160.5	11.93	136.4	18.60	5.05	162.4	0.95
	Negative	50	1.80	85.4	3.56	149.1	12.04	126.7	18.63			

Notes: Δ_cr_ is the corresponding displacement when the loading reaches crack load (*P*_cr_); Δ_y_ is the yield displacement when the force reaches yield load (*P*_y_); *P*_m_ and *P*_u_ are the peak load and ultimate load, respectively. Δ_m_ and Δ_u_ are the peak displacement and ultimate displacement, respectively. Meanwhile, *µ* represents the ductility factor, and the *P*_am_ represents the calculating results for the shear capacity of PC walls by the Equation (9).

## Data Availability

The data presented in this study are available on request from the corresponding author.

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
