# Peer review of "Experimental Study on Seismic Behavior of PC Walls with Alveolar-Type Horizontal Joint under Pseudo-Static Loading"

_materials, 2022, doi:10.3390/ma15062301_

Round 1

Reviewer 1 Report

Review of the paper: Seismic behavior on precast concrete walls with alveolar-type 2 horizontal joint under pseudo-static loading

The authors propose a new connection system for the horizontal joints on precast concrete (PC) wall-panel structures constituted by an ‘alveolar’ horizontal joint and two vertical grouted sleeve connections. The seismic behaviour of the walls was tested with an experimental campaign by carrying out pseudo-static loading tests on 6 specimens.

The tackled issue is relevant; however, some issues about the interpretation of the results may be outlined. In addition, English should be checked by a native speaker, especially for what concern the introduction. Some major observations are:

  1. Introduction: English should be corrected - e.g. typo/English errors are present in lines 48 and 54 (‘as’ instead of ‘is’), 59, 64 (‘could be ensure’), 70-71 (‘it is still need to pour concrete or wet operation on site after the wall-panels are assembled’), 72 (‘needs to be connected connections and low tolerances’), among others.
  2. Line 114: the proposed connection should be better explained. How is it made? (explanation in lines 114-115: ‘a novel alveolar-type horizontal joint connections with high at both ends and low in the middle,’ is not clear). Which material is adopted for the connection? In Line 140 it is stated that is a ‘joint made of mortar’, in line 181 it is stated there is ‘a plastic film with much micro-holes’. A better description accompanied by a figure would clarify. It is not clear why the connection is named ‘alveolar’ too.
  3. Lines 117-119: ‘Meanwhile, the horizontal joint was weak connected by several single-row steel grouted sleeve connectors to realize the indirect connection between the vertical walls,’. Why is it considered an indirect connection?
  4. Line 140: what mean the authors when say that the bottom beam is made of mortar?
  5. Lines 146-148: ‘Its connection strength was approximately of 40% based on design of structural-performance requirements’. Please clarify this sentence, 40% of what?
  6. Line 160: the axial compression ratio and the shear span ratio should be defined.
  7. Lines 166-170: please clarify
  8. Table 1: are the selected axial compression ratios representative of actual load configuration real buildings? May exist a case in which a load bearing wall has 0 vertical loading? Is 0.2 the maximum axial compression ratio which could be expected in a real building? Or these are just illustrative numbers that do not represent real cases? Please clarify
  9. Line 212: how did the authors manage to keep the axial load constant during the test? The specimen tends to tilt during the test, loading one side and unloading the other. Was this taken into account in some way? In addition, in real buildings, the top of the wall would have been connected to a floor, which would restrain and limit the rotation of the panel. Please add a comment about it.
  10. Figure 4: please define T. in addition, a sliding in correspondence of the interface could have been expected, why the authors did not put any instrument along the interface to better control both the sliding and the rotation? Please add a comment about it.
  11. Section 3 and 4: the expected mechanism is quite clear, as also explained in section 4.3: once the friction resistance (which is clearly function of both the area at the interface and the axial force) is overcome, the block on the top start to slip and to rotate. The rotation will create an overload at the corner which cause the collapse of the block. This physical mechanism is not well described, and a more muddled and long explanation of the mechanism is provided.
  12. Section 3 and 4.1: a reference to the rocking walls would have been interesting
  13. Figure 6: being cyclic tests, it would have been interesting to show, at least for one specimen, the evolution of the cracks during the various loading cycles
  14. Line 423-5: there is not a reduction of the lateral stiffness, it is also clear from the curves of Figure 9; the friction load is simply overcome.
  15. Lines 536-538: ‘However, as the axial compression ratio increases continuously, these cracks will close immediately when the specimen after cracked subjected to repeated loading due to the larger axial force.’ Please clarify
  16. Section 4.5: which displacement is plotted? What do the authors mean with ‘displacement load’?
  17. Conclusions: the behavior of the specimens could have been expected, the axial compression ratio and the thickness influence the response because directly influence the friction resistance at the interface; the grouted connections are the only elements that contrast the sliding of the block and increase their bending capacity, so it could have been expected they had a positive impact to the wall response. The representativeness of the specimens and of the loading conditions of a real building should also be discussed.

Author Response

Please see the attachment, thank you!

Reviewer 2 Report

This paper presents the results of an experimental study on precast concrete walls with alveolar-type horizontal joint connections. The effects of axial load ratio, wall thickness, and the presence of sleeve connection at the horizontal joint were investigated experimentally. The subject of this paper is interesting and the presented results can be used by practice engineers and used to validate numerical models. In general, the paper can be of importance to those working in the field as it contains interesting information. Before any further consideration of acceptance, there are still some problems and suggestions in this manuscript.

1 - The authors have only included the compressive strength of the mortar used in alveolar-type connections. They need to provide all material properties of the mortar including compressive and tensile strengths, the elastic modulus, and Poisson’s ratio.

2 - The authors must advise how the size effect can affect the results presented in the manuscript. The small specimens used in this study are not representative of the real wall specimens used in building construction so it should be clarified whether the conclusions are still valid for real cases.

3 - The values of the parameters of the proposed predictive equation for the shear strength of precast concrete walls are assumed with no detailed discussion. It is recommended to provide the source of values assigned to the equation parameters.

4 - The proposed equation for the shear capacity of PC walls can not be generally used for strength estimation of these walls as it has been developed based on the results of a limited number of test specimens. The authors have to conduct a comprehensive parametric study using a validated FEM to investigate the effects of various design variables with a wide range of values. For example, a key design variable might be the number of added bars connecting the two wall panels. The effects of some important design variables (added bar properties and numbers, mortar properties, etc.) are missed.

Author Response

Please see the attachment, thank you!

Reviewer 3 Report

This paper presents a novel design approach for precast concrete walls for earthquake resistance.

This paper is well written and is useful to both practicing engineers and research community. 

Suggestion:

The authors should address the essence of wall height on the system stability under seismic loadings.

Round 2

Reviewer 1 Report

The authors replied satisfactorily to all the previous observations.